# EGF-Enhanced GnRH-II Regulation in Decidual Stromal Cell Motility through Twist and N-Cadherin Signaling

**DOI:** 10.3390/ijms242015271

**Published:** 2023-10-17

**Authors:** Hsien-Ming Wu, Liang-Hsuan Chen, Hong-Yuan Huang, Hsin-Shih Wang, Chia-Lung Tsai

**Affiliations:** Department of Obstetrics and Gynecology, Chang Gung Memorial Hospital Linkou Medical Center, School of Medicine, Chang Gung University, Taoyuan 333, Taiwan; elsachen0119@gmail.com (L.-H.C.); hongyuan@cgmh.org.tw (H.-Y.H.); hswang0707@gmail.com (H.-S.W.); cltsai@cgmh.org.tw (C.-L.T.)

**Keywords:** EGF, GnRH-II, decidua, motility, Twist, N-cadherin

## Abstract

Crucial roles in embryo implantation and placentation in humans include the invasion of the maternal decidua by extravillous trophoblasts and the motile behavior of decidual endometrial stromal cells. The effects of the epidermal growth factor (EGF) and GnRH-II in the endometrium take part in early pregnancy. In the present study, we demonstrated the coaction of EGF- and GnRH-II-promoted motility of human decidual endometrial stromal cells, indicating the possible roles of EGF and GnRH-II in embryo implantation and early pregnancy. After obtaining informed consent, we obtained human decidual endometrial stromal cells from decidual tissues from normal pregnancies at 6 to 12 weeks of gestation in healthy women undergoing suction dilation and curettage. Cell motility was evaluated with invasion and migration assays. The mechanisms of EGF and GnRH-II were performed using real-time PCR and immunoblot analysis. The results showed that human decidual tissue and stromal cells expressed the EGF and GnRH-I receptors. GnRH-II-mediated cell motility was enhanced by EGF and was suppressed by the knockdown of the endogenous GnRH-I receptor and EGF receptor with siRNA, revealing that GnRH-II promoted the cell motility of human decidual endometrial stromal cells through the GnRH-I receptor and the activation of Twist and N-cadherin signaling. This new concept regarding the coaction of EGF- and GnRH-promoted cell motility suggests that EGF and GnRH-II potentially affect embryo implantation and the decidual programming of human pregnancy.

## 1. Introduction

Embryo implantation happens in the decidualization of the uterine endometrium. Endometrial stromal cells are significant in the decidual programming of human pregnancy. Transforming endometrial stromal cells into decidual stromal cells is important in embryo implantation [1,2]. Extravillous trophoblast invasion of the maternal decidua is essential for human embryo implantation and placentation. Decidual stromal cells’ migratory and invasive potentials are involved in embryo implantation and the programming of human pregnancy [3,4]. Despite the mobile and invasive abilities of decidual stromal cells being known, their roles in the mechanism of embryo implantation and the processes of pregnancy remain unclear. The hypothalamic–pituitary–ovary hormone axis is the key element in the control of steroidogenesis and the subsequent endometrial environment for embryo implantation. However, the underlying mechanisms of the action of the hypothalamic–pituitary–ovary hormone axis in an endometrial environment for embryo implantation are still areas in which there are gaps in our knowledge.

GnRH, the hypothalamic peptide, preserves reproductive tissues and maintains their functions [5,6,7]. Compared with GnRH-I, GnRH-II exists more commonly in peripheral tissues, indicating that GnRH-II may have additional roles in mammals. A previous study showed that GnRH-II directly affects endometrial cancer cell growth [8]. These findings suggested that GnRH-II mediates the behavior of endometrial cells. Additionally, epidermal growth factor (EGF), an autocrine growth factor, plays key roles in cell biology, including cell growth, proliferation, adhesion, and motility, and is a critical therapeutic target in reproductive cells [9,10]. A previous study has shown that GnRH-II expression is regulated by EGF activation of its receptor and several intracellular signaling pathways in gynecological cells [11]. Furthermore, the expression of GnRH-II regulates gynecological cell motility in a particular autocrine/paracrine EGF-stimulated method. The coaction of EGF and GnRH-II in decidual endometrial stromal cells (DSCs) is not well established, and the mechanism related to embryo implantation and early pregnancy by which EGF and GnRH-II regulate the motility of decidual stromal cells remains unknown. Twist, a helix–loop–helix transcription factor, participates in cell invasion and migration by regulating the expression of N-cadherin in cells [12,13,14]. N-cadherin, a member of the superfamily of integral membrane glycoproteins, modulates cell adhesion and induces cell motility during disease progression [15,16,17]. Twist and N-cadherin signaling pathways can regulate cell motility and tissue remodeling in several models, and the expression of Twist and N-cadherin was reported to play an important role in epithelial–mesenchymal transition (EMT) events [18]. In this study, we investigated the effects of EGF and GnRH-II on the motility of decidual stromal cells and the coaction mechanisms involved. Exploring cotreatment with EGF and GnRH-II, the present results offer a potential target in embryo implantation and placentation in humans.

## 2. Results

### 2.1. GnRH-II Induces Human Decidual Endometrial Stromal Cell Motility

To study the functional role of GnRH-II in human decidual stromal cell migration and invasion, we evaluated in vitro cell motility following GnRH-II agonist stimulus using Transwell migration and invasion assays. The GnRH-II agonist induced decidual stromal cell migration through the uncoated porous filter in a dose-dependent pattern at 1 nM to 1 μM concentrations, with the most significant effect at 1 μM (Figure 1A). The GnRH-II agonist stimulated the invasion of decidual stromal cells through the Matrigel-coated filter in a dose-dependent mode at 1 nM to 1 μM concentrations, with the most prominent phenomenon at 1 μM (Figure 1B).

### 2.2. Expression of the GnRH-I Receptor (GnRH-IR) in Decidual Stromal Cells

To identify GnRH-IR expression, we extracted the primary culture of human decidual endometrial stromal cells and investigated the GnRH-IR expression using immunoblot analysis and immunohistochemical studies [19,20]. Our previous studies confirmed that GnRH-IR was present in human decidual endometrial stromal cells purified from individual patients [19,20].

### 2.3. The GnRH-II-Stimulated Migration and Invasion of Human Decidual Endometrial Stromal Cells through GnRH-I Receptors

To confirm whether the effects of GnRH-II on cell motility were interceded by the GnRH-I receptor, we used the GnRH-I receptor siRNA to transfect human decidual endometrial stromal cells, inhibiting endogenous GnRH-I receptor expression. The 50 nM GnRH-I receptor siRNA challenge reduced the GnRH-I receptor protein expression (Figure 2A) in human decidual endometrial stromal cells. In addition, the down-regulated endogenous GnRH-I receptor remarkably suppressed GnRH-II-stimulated cell migration (Figure 2B) and decreased GnRH-II-promoted cell invasion (Figure 2C). These data revealed that the GnRH-I receptor moderated GnRH-II-induced cell migration and invasion in human decidual endometrial stromal cells.

### 2.4. Expression of EGFR in Decidual Endometrial Tissue and Stromal Cells

To investigate the expression of EGFR, we digested the primary culture of human decidual endometrial stromal cells and examined the expression of EGFR mRNA using RT-PCR analysis. EGFR was identified with RT-PCR in human decidual endometrial stromal cells isolated from five individuals (Figure 3A). Ishikawa endometrial cancer cells played the role of a positive control.

Immunohistochemical studies revealed that EGFR was expressed in the human decidual endometrial tissues and stromal cells (Figure 3B).

### 2.5. GnRH-II and EGF Cotreatment to Enhance the Motility of Decidual Stromal Cells

To examine the effect of EGF-enhanced GnRH-II regulation on decidual stromal cell motility, human decidual endometrial stromal cells were individually treated with 100 ng/mL EGF or 100 nM GnRH-II or in combination for 24 h before the migration assay and invasion assay. We report that individually EGF-treated or GnRH-II-treated cells reveal increased effects of migration and invasion compared with untreated controls (Figure 4A,B). Furthermore, to assess the possibility that GnRH-II functions together with EGF to promote the cell migration and invasion of human decidual endometrial stromal cells, we simultaneously manipulated the cells with EGF and GnRH-II, and the results show that EGF and GnRH-II have an add-on action on the migration and invasion of human decidual endometrial stromal cells (Figure 4A,B).

### 2.6. Impacts of Human EGF Receptor (EGFR) siRNA Transfection on Decidual Stromal Cells

To investigate whether EGFR intermediated in the impacts of EGF on cell migration and invasion, we transfected human decidual endometrial stromal cells with an EGFR siRNA to suppress the endogenous EGFR expression. The results showed down-regulated EGFR protein expression (Figure 5A) and mRNA expression (Figure 5B) after treatment with 50 nM EGFR siRNA for 24 h and 48 h in human decidual endometrial stromal cells. In addition, the endogenous EGFR knockdown significantly reduced EGF-induced cell migration (Figure 5C) and decreased EGF-stimulated cell invasion (Figure 5D). Collectedly, our experiments revealed that EGFR intermediates EGF-stimulated cell migration and invasion in human decidual endometrial stromal cells.

### 2.7. EGF Stimulates GnRH-I Receptors and GnRH-II Expression in Human Decidual Endometrial Stromal Cells

To elucidate the consequences of EGF-induced expression of GnRH-I receptors and GnRH-II in human decidual endometrial stromal cells, we treated decidual stromal cells with 100 ng/mL EGF for 24 h before RT-PCR and immunoblot analysis. We demonstrated that EGF-treated cells increased the mRNA expression of GnRH-I receptors (Figure 6A) and GnRH-II (Figure 6B) compared with untreated controls. We also report that EGF-treated cells enhanced the protein expression of GnRH-I receptors compared with untreated controls (Figure 6C). Knockdown of the endogenous EGFR with 50 nM EGFR siRNA treatment significantly suppressed the mRNA expression of EGF-induced GnRH-I receptors (Figure 6D) and GnRH-II (Figure 6E). Our experiments demonstrated that EGFR mediates the EGF-induced expression of GnRH-I receptors and GnRH-II in human decidual endometrial stromal cells.

### 2.8. GnRH-II-Induced Twist and N-Cadherin Expression on the Cell Motility of Human Decidual Endometrial Stromal Cells

Twist and N-cadherin are initially reported to induce cell motility and angiogenesis. To investigate whether Twist and N-cadherin participated in the GnRH-II-promoted cell motility of decidual endometrial stromal cells, we applied an immunoblot analysis to measure the expression of Twist and N-cadherin in GnRH-II-treated decidual stromal cells. With treatment concentrations of 1 nM to 10 μM, GnRH-II increased the protein expression of Twist and N-cadherin (Figure 7A). Furthermore, we isolated total RNA from decidual stromal cells with treatments of 1 nM to 10 μM GnRH-II compared with untreated cells. We obtained cDNA with RT-PCR to demonstrate the impact of GnRH-II on Twist and N-cadherin mRNA expression (Figure 7A). A challenge with 50 nM GnRH-I receptor siRNA suppressed GnRH-I receptor protein expression in human decidual endometrial stromal cells (Figure 7B). In addition, the down-regulated endogenous GnRH-I receptor remarkably decreased the mRNA and protein expression of GnRH-II-induced Twist and N-cadherin (Figure 7B). These data show that the GnRH-II-induced expression of Twist and N-cadherin in human decidual endometrial stromal cells is mediated by GnRH-I receptors.

### 2.9. Twist and N-Cadherin Regulate GnRH-II-Induced Cell Motility in Human Decidual Endometrial Stromal Cells

To elucidate whether Twist and N-cadherin mediate cell invasion and migration by GnRH-II, we transfected human decidual endometrial stromal cells with Twist siRNA and N-cadherin siRNA individually to down-regulate endogenous Twist and N-cadherin expression. Twist expression was knocked down in decidual stromal cells treated with 50 nM Twist siRNA (Figure 8A). The down-regulated endogenous Twist significantly abolished GnRH-II-induced cell invasion and migration (Figure 8A). N-cadherin expression was down-regulated in decidual endometrial stromal cells treated with 50 nM N-cadherin siRNA. The down-regulated endogenous N-cadherin significantly suppressed GnRH-II-stimulated cell invasion and migration (Figure 8B).

### 2.10. Conclusions

These data demonstrate that Twist and N-cadherin regulate GnRH-II-induced cell invasion and migration in decidual stromal cells. Throughout the present study, the signaling pathways of decidual stromal cell motility were established via cotreatment with GnRH-II and EGF (Figure 9).

## 3. Discussion

The GnRH pathway plays a critical role in the hypothalamus–pituitary–gonadal axis of reproduction [21]. The excess production of GnRH-induced ovarian estrogen and the hypersensitive receptors of peptides and steroids result in uterine diseases, including leiomyoma, endometrial cancer, and endometriosis [22,23]. Compared with GnRH-I, GnRH-II has more potent effects on extra-pituitary tissues, indicating a possible modulator in the endometrial environment. Regulations between the embryo and the stroma provide embryonic trophoblast invasion into decidual stromal cells during human pregnancy [24,25,26].

The ability of human decidual endometrial stromal cells to move away from the implantation site smooths the process of trophoblast invasion into the stromal compartment. Embryo implantation would be disturbed by impaired function of human decidual endometrial stromal cell motility. Consequently, decidual stromal cell motility may regulate the mechanism of successful embryo implantation. The GnRH-I receptor, a member of the GPCR family, transfers the signals through multiple G protein subunits formed by seven transmembrane domains and participates in numerous signaling pathways [27]. Besides GPCRs, classical growth factors, such as EGF, also act in differentiation, proliferation, and cellular transformation [28,29]. The well-recognized level of epigenetic regulation is executed by some noncoding RNAs [30], which have a role in the expression of many genes that participate in macrophage polarization and tissue remodeling. Some studies report that cellulose nanofiber exposure stimulated dynamic changes in the miRNA transcript [30], indicating that differentially expressed miRNA is involved in several different signaling pathways, such as EGF, GnRH-R, integrin, and other adhesion molecule signaling, which are essential for inflammation and tissue remodeling during human pregnancy [25]. In previous studies, our group demonstrated that GnRH-II expression is regulated by EGF activation of its receptor and several intracellular signaling pathways in gynecological cells [11]. Meanwhile, there is evidence that EGF-stimulated GnRH-II expression assembles a particular autocrine/paracrine regulation in gynecological cell motility. The molecular pathways of GnRH-II and EGF involved in decidual stromal cell motility remain unclear. The effects of both GnRH-I and GnRH-II on GnRH-I receptors are inconclusive. The previous report proved that proteolytic release of local EGF-like ligands from transmembrane precursors regulates EGFR via GPCRs [31]. In the present study, for the first time, we demonstrated that cotreatment with EGF and GnRH-II modulates the invasion and migration of human decidual endometrial stromal cells by activating Twist and N-cadherin pathways through the EGF receptor and GnRH-I receptor, providing an insight into the prospect of embryo implantation in early pregnancy.

GnRH analogs and its receptor regulate cell growth in human decidual endometrial stromal cells [19]. In this study, we discovered significant effects on decidual stromal cell invasion and migration after EGF and GnRH-II treatment. The results support that EGF and GnRH-II directly promote decidual stromal cell motility and induce the invasion and migration of human decidual endometrial stromal cells. Previous studies [32,33,34,35,36,37,38] also reported that EGF and GnRH-II mediate reproductive cell motility and behavior. According to the emerging evidence, expression of the EGF receptor and GnRH-I receptor contributes to the cell type-specific signaling of EGF and GnRH-II on the motile machinery within cells.

This study used GnRH-I receptor siRNA to down-regulate the protein expression of GnRH-I receptors in human decidual endometrial stromal cells [19,20]. Knockdown of GnRH-I receptors suppressed GnRH-II-stimulated decidual stromal cell invasion and migration, indicating that GnRH-II acted in decidual endometrial stromal cells depending upon GnRH-I receptors. Previous studies [8,39,40,41] have shared the similar concept that the GnRH-I receptors commonly regulate the effects of both GnRH-I and GnRH-II in gynecologic cells and endometrial cells, which is related to embryo implantation and pregnancy. On the other hand, in order to explore the effects of EGF in endometrial cells, our study used EGF receptor siRNA to down-regulate the protein expression of EGF receptors in human decidual endometrial stromal cells. Knockdown of receptors suppressed EGF-stimulated decidual stromal cell invasion and migration, revealing that EGF acted in decidual stromal cells depending upon EGF receptors, indicating that EGF may play a role in regulating the motility of endometrial stromal cells, which is associated with embryo implantation and pregnancy.

Twist and N-cadherin signaling pathways determine a cell’s behavior, mediated by different extracellular stimuli. Epithelial–mesenchymal transition (EMT) is a physiological process, whereas epithelial cells lose adhesive properties and gain some invasive mesenchymal features with further production of the extracellular matrix (ECM) [42,43]. EMT development is associated with cancer biology, injury, fibrosis, healing, and tissue remodeling [44,45]. More importantly, EMT can be investigated by screening biomarkers that reflect the loss of epithelial characteristics and acquired mesenchymal properties, such as gains in Twist and N-cadherin [46]. Twist and N-cadherin signaling pathways can modulate EMT in rat models, and the expression of Twist and N-cadherin genes was found to endure in tumor cells with stem cell characteristics and play an important role in EMT events [18]. To elucidate the signaling of EGF, GnRH-II, Twist, and N-cadherin action in endometrial stromal cells, we identified the relationship between the EGF, GnRH-II, Twist, and N-cadherin signaling pathways to provide a better understanding of the crosstalk between EGF and GnRH-II. Identifying the relationship between the EGF, GnRH-II, Twist, and N-cadherin signaling pathways provides a better understanding of the crosstalk between EGF and GnRH-II. It was observed in our study that GnRH-II (1 μM) activated Twist and N-cadherin in decidual stromal cells. Pretreatment with GnRHR siRNA in decidual stromal cells markedly attenuated the expression of Twist and N-cadherin, indicating that GnRH-II regulates the decidual stromal cells through Twist and N-cadherin signaling. Moreover, pretreatment with Twist and N-cadherin siRNA reduced GnRH-II-stimulated cell invasion and migration, showing that the GnRH-II-induced cell motility in decidual stromal cells is activated through Twist and N-cadherin signaling pathways. Taken together, we demonstrated that the Twist and N-cadherin pathways regulate GnRH-II-induced decidual stromal cell motility. Targeting Twist and N-cadherin signaling manipulates cell invasion and migration abilities and allows for exploring the impacts of GnRH-II on decidual stromal cells during embryo implantation.

Proteolysis, increased cell migration, and decreased cell adhesion contribute to cell motility. EGF is a common factor involved in cell motility processes, such as cancer metastasis. Our study demonstrated that cotreatment with EGF and GnRH-II activated the cell invasion and migration of decidual stromal cells through up-regulation of the Twist and N-cadherin signaling pathways. These novel findings were proved by the suppressive effects on GnRH-II-stimulated cell motility when blocking Twist and N-cadherin expression by siRNA.

## 4. Materials and Methods

### 4.1. Cell Culture, Isolation, and Identification

To understand the interaction of EGF and GnRH-II in human decidual stromal cells, human decidual stromal cells were collected from decidual tissue of healthy women aged 25–35 years undergoing elective termination of a normal gestation at 6–12 weeks after informed consent was obtained. The Institutional Review Board of Chang Gung Memorial Hospital approved the human subjects’ involvement in this study (CGMH-IRB numbers 98-3589B, 99-3396C, and 100-3879C). Using a modified protocol, we retrieved human decidual stromal cells from decidual tissue through enzymatic digestion and mechanical dissociation [19,20,47]. We minced and treated the decidual tissue with type IV collagenase (Sigma-Aldrich, St. Louis, MO, USA) and DNase type I in a shaking water bath at 37 °C for 90 min. The digested cells passed through a 70 μm filter. We gathered the decidual stromal and epithelial cells in a 50 mL tube and isolated decidual stromal cells from epithelial cells with a 45 μm filter. After centrifugation at 1000× *g* for 5 min at room temperature, the decidual stromal cell pellets were collected, washed once, resuspended, and plated in DMEM containing 25 mM glucose, L-glutamine, and antibiotics (100 U/mL penicillin and 100 μg/mL streptomycin), and supplemented with 10% fetal bovine serum. In order to confirm the purity of the primary decidual stromal cells, the cells were seeded into a 24-well culture plate with a density of 1 × 10^4^ cells/mL. After achieving 80% confluence, the cells were washed with PBS twice. Secondly, 1 mL ice-cold 4% paraformaldehyde was added into each well to fix the cells for 20 min on ice. After this, the fixed cells were washed with PBS and stored at 4 °C in PBS for immunofluorescence recognition with vimentin and cytokeratin 7 (CK7) (Appendix A).

### 4.2. Reagents

We purchased the GnRH-II agonist (D-Arg6, AzaGly10-GnRH II), a synthetic decapeptide, from Bachem (San Carlos, CA, USA). We acquired human EGF from Sigma Chemical Co. (St. Louis, MO, USA).

### 4.3. Immunoblot Analysis

The cells were lysed in buffer containing 20 mM Tris, pH 7.4, 2 mM EGTA, 2 mM Na_2_VO_3_, 2 mM Na_4_P_2_O_7_, 2% SDS, 2% Triton X-100, 1 μM leupeptin, 1 μM aprotinin, and 1 mM PMSF. We used a protein assay kit by BSA standards to determine the protein concentration according to the manufacturer’s instructions (Bio-Rad Laboratories, Hercules, CA, USA). The amounts of cell lysate were distributed equally via SDS polyacrylamide gel electrophoresis (PAGE) and transferred to a nitrocellulose membrane (Hybond-C, Amersham Pharmacia Biotech Inc., Oakville, ON, USA). After blocking with Tris-buffered saline (TBS) containing 5% nonfat dry milk for 1 h, we incubated the membranes at 4 °C overnight, with anti-GnRH-I receptor (Neomarker, Fremont, CA, USA), polyclonal total EGFR antibody (Cell Signaling Technology, Danvers, MA, USA), and anti-N-cadherin (Millipore, Darmstadt, Germany) or anti-Twist (Thermal, MA, USA) antibody, followed by incubating them with HRP-conjugated secondary antibody. We read the immunoreactive bands with an enhanced chemiluminescence (ECL) kit. The membrane was stripped by stripping buffer (62.5 mM Tris, 10 mM DTT, and 2% SDS, pH 6.7) at 50 °C for 30 min and re-probed with anti-β-actin antibody (Santa Cruz, Dallas, TX, USA) as a loading control.

### 4.4. Immunohistochemistry (IHC)

We performed immunohistochemistry (IHC) on sections of human decidual endometrial tissue to identify EGF receptor protein expression as per previously reported methods [18]. Four-micrometer-thick formalin-fixed, paraffin-embedded (FFPE) tissue sections were deparaffinized in xylene and rehydrated with a graded series of ethanol solutions. We stained the tissue sections with an antihuman EGF receptor polyclonal antibody (Cell Signaling Technology; 1:100) using an automated IHC stainer with the Ventana Basic DAB Detection kit (Tucson, AZ, USA) according to the manufacturer’s protocol. We performed counterstaining with hematoxylin. The tissue section without EGF receptor antibody staining served as a negative control.

### 4.5. Small Interfering RNA Transfection

We obtained siGENOME ON-TARGETplus SMARTpool human GnRH-I receptor siRNA, human EGF siRNA, human Twist siRNA, human N-cadherin siRNA, and siCONTROL NON-TARGETINGpool siRNA from Dharmacon (Lafayette, CO, USA). We transfected the cells with siRNA (100 nM) using Lipofectamine RNAiMAX. After a 24 h transfection, the medium was removed and changed to a fresh serum-free medium. Cells were transfected with 100 nM si-GLO (Dharmacon) for 24 h, and the siRNA transfection efficiency was verified via fluorescent microscopy.

### 4.6. Invasion and Migration Assays

We operated invasion and migration assays in Boyden chambers with minor modifications. Cell culture inserts (24-well, pore size 8 μm; BD Biosciences, Mississauga, ON, USA) were seeded with 1 × 10^5^ cells in 250 μL of medium with 0.1% FBS. Inserts precoated with growth factor-reduced Matrigel (40 μL, 1 mg/mL; BD Biosciences) were applied for invasion assays, whereas uncoated inserts were used for migration assays. Medium with 10% FBS (750 μL) was placed in the lower chamber as a chemotactic agent. After incubation for 48 h (invasion) or 24 h (migration), we wiped non-invading/migrating cells from the upper side of the membranes and fixed cells on the lower side of the membranes in cold methanol (−20 °C); this was followed by an air drying procedure. The cells that had not invaded the filter were removed via wiping, and the cells that had penetrated the lower surface of the filter were fixed with ice-cold methanol and stained with 0.5% crystal violet.

### 4.7. Real-Time PCR

After treatment with EGF, we removed the medium from the culture dish. We extracted total RNA from cultured decidual stromal cells using TRIzol reagent (Invitrogen, Waltham, MA, USA). We evaluated the total RNA concentration with A260/280 nm spectrophotometric analysis. Complementary DNA (cDNA) was synthesized from RNA samples using SuperScript III First-strand Synthesis System (Invitrogen). Gene expression was assessed using Bio-Rad iCycler (iQ5 Real-Time PCR Detection System, Bio-Rad Laboratories, Inc., Foster City, CA, USA) with Applied Biosystems™ Power SYBR™ Green PCR Master Mix. Expression was normalized relative to the housekeeping gene GAPDH. Following the manufacturer’s procedure, primer sequences for EGFR used the following primer pair: sense, 5′-AGCTTTGCAGCCCATTTCTA-3′ and antisense, 5′-CAGCGC TACCTTGTCATTCA-3′. The primers for GnRH-II are sense, 5′-TCTGTTCCCCTCCAACTTTCTTC-3′ and antisense, 5′-AGGTCCATCCATCTTTCCTTCA-3′. The primers for GnRHR are sense, 5′-ACCGCTCCCTGGCTATCAC-3′ and antisense, 5′-ACTGTTCCGACTTTGCTGTTGCT-3′. The primers for Twist are sense, 5′-AGTCC GCAGT CTTAC GAGGA-3′ and antisense, 5′-GCAGAGGTGT GAGGA TGGT-3′. The primers for N-cadherin are sense, 5′-GCGTCTGTAGAGGCTTCTGG-3′ and antisense, 5′-GCCACTTGCCACTTTTCCTG-3′. The primers for glyceraldehyde-3-phosphate dehydrogenase (GAPDH) are sense, 5′-GAGTCAACGGATTTGGTCGT-3′ and antisense, 5′-GACAAGCTTCCCCGTTCTAG-3′. All real-time experiments were run in triplicate, and a mean value was used to determine mRNA levels. Negative controls containing water instead of sample cDNA were used in each experiment. Relative quantification of the mRNA levels for EGFR in endometrial stromal cells was evaluated using the comparative CT method with glyceraldehyde-3-phosphate dehydrogenase as an endogenous control and with the formula 2^−△△Ct^.

### 4.8. Statistical Analysis

The results are presented as the means ± SEM. We conducted a statistical evaluation with a *t*-test for paired data. Multiple comparisons were first analyzed via one-way ANOVA, followed by Tukey’s multiple comparison test. A significant difference was defined as *p* < 0.05.

## 5. Conclusions

We proved the mechanism of embryo implantation in which the EGF and GnRH-II binding of EGF receptors and GnRH-I receptors activates Twist and N-cadherin pathways, contributing to decidual stromal cell motility. The results pointed out that EGF and GnRH-II, as well as other molecules that regulate Twist and N-cadherin expression in decidualized endometrium, are potential therapeutic targets for embryo implantation failure in the treatment of infertility. Additionally, the newly discovered mechanisms provide us with a better understanding of embryo implantation and the decidual programming of human pregnancy.

## Figures and Tables

**Figure 1 ijms-24-15271-f001:**
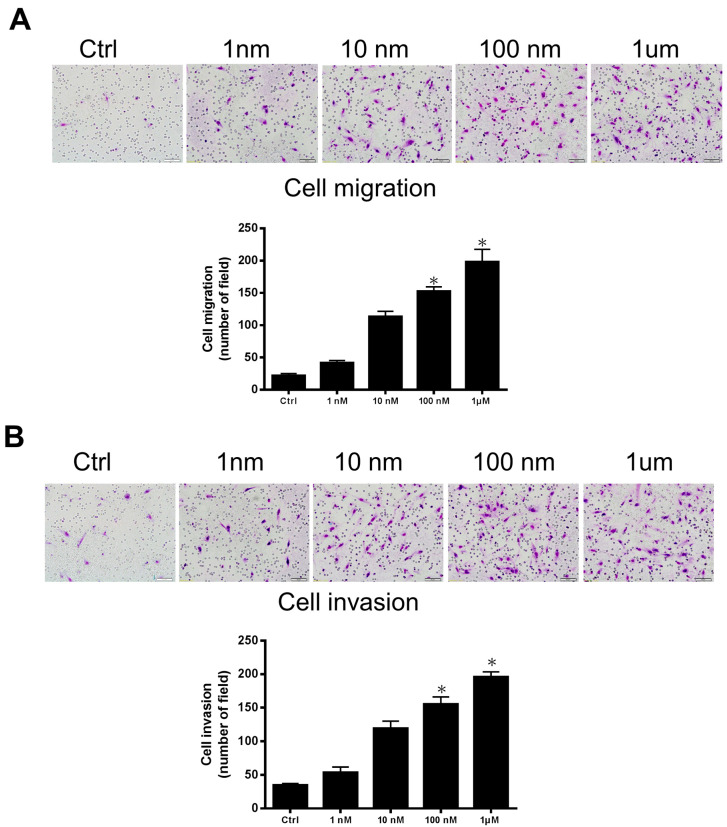
GnRH-II induces decidual stromal cell motility. (**A**) Human decidual endometrial stromal cells were seeded by applying the Transwell migration assay. The GnRH-II agonist promoted decidual stromal cell migration through the uncoated porous filter in a dose-dependent pattern at 1 nM to 1 μM concentrations, with the most significant effect at 1 μM. (**B**) Decidual endometrial stromal cells were seeded onto a Matrigel-precoated filter in the Transwell chambers with or without the GnRH-II agonist (1 nM to 1 μM concentrations, as indicated). After culturing for 24 h (migration) and 48 h (invasion), we removed cells in the upper side of the filter and fixed, stained, and counted the migrated or invaded cells. Low, representative pictures. Columns, the mean number of migrated or invaded cells of five field triplicate wells from three independent experiments; bars, SE; * *p* < 0.05, versus control. Scale bars represent 50 μm.

**Figure 2 ijms-24-15271-f002:**
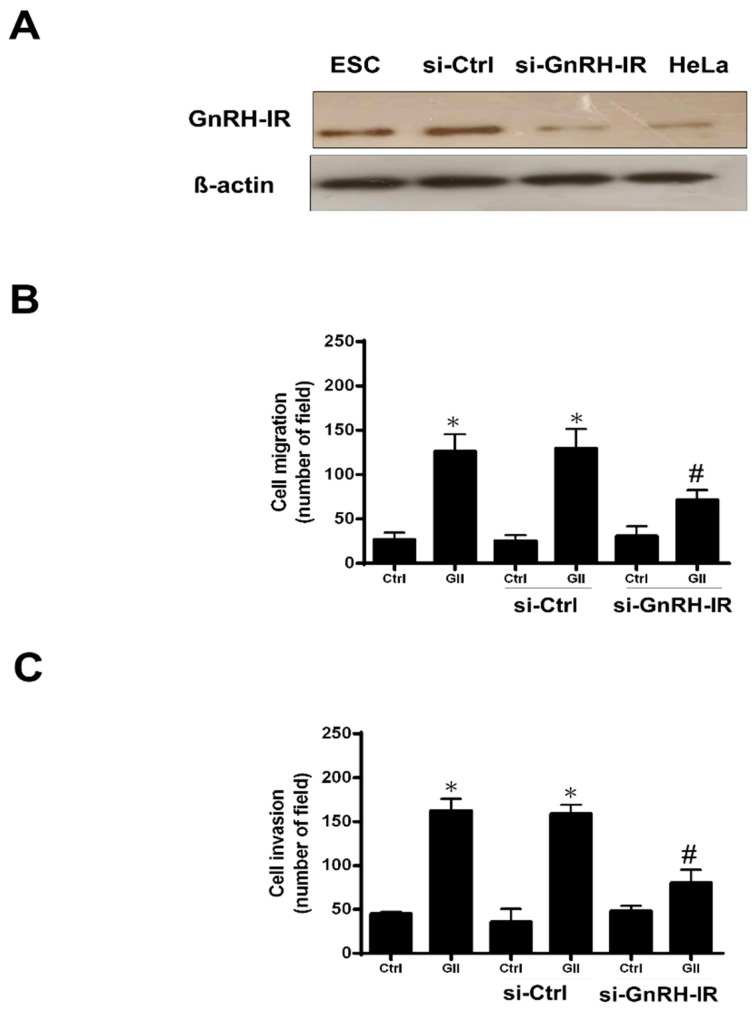
The GnRH-II-stimulated migration and invasion of decidual endometrial stromal cells (ESC) through GnRH-I receptors in humans. (**A**) Immunoblot assay was applied to identify GnRH-I receptor levels in human decidual endometrial stromal cells. Human decidual endometrial stromal cells were transfected by human si-GnRH-IR or scrambled siRNA (si-Ctrl) with Lipofectamine RNAiMAX for one day. Negative control was compared by HeLa cells. (**B**) The impacts of si-GnRH-IR transfection on GnRH-II-stimulated cell migration. GnRH-II (1 μM) was used to treat the pre-transfected cells with si-GnRH-IR for 24 h. A migration assay was applied to evaluate cell motility. The results are shown as the mean ± SEM from three individual experiments (* *p* < 0.05, versus control; # *p* < 0.05, versus GnRH- II). (**C**) The impacts of si-GnRH-IR transfection on GnRH-II-induced cell invasion. GnRH-II (1 μM) was used to treat the pre-transfected cells with si-GnRH-IR for 48 h. The invasion assay was utilized to assess cell motility. The results are shown as the mean ± SEM from three individual experiments (* *p* < 0.05, versus control; # *p* < 0.05, versus GnRH-II).

**Figure 3 ijms-24-15271-f003:**
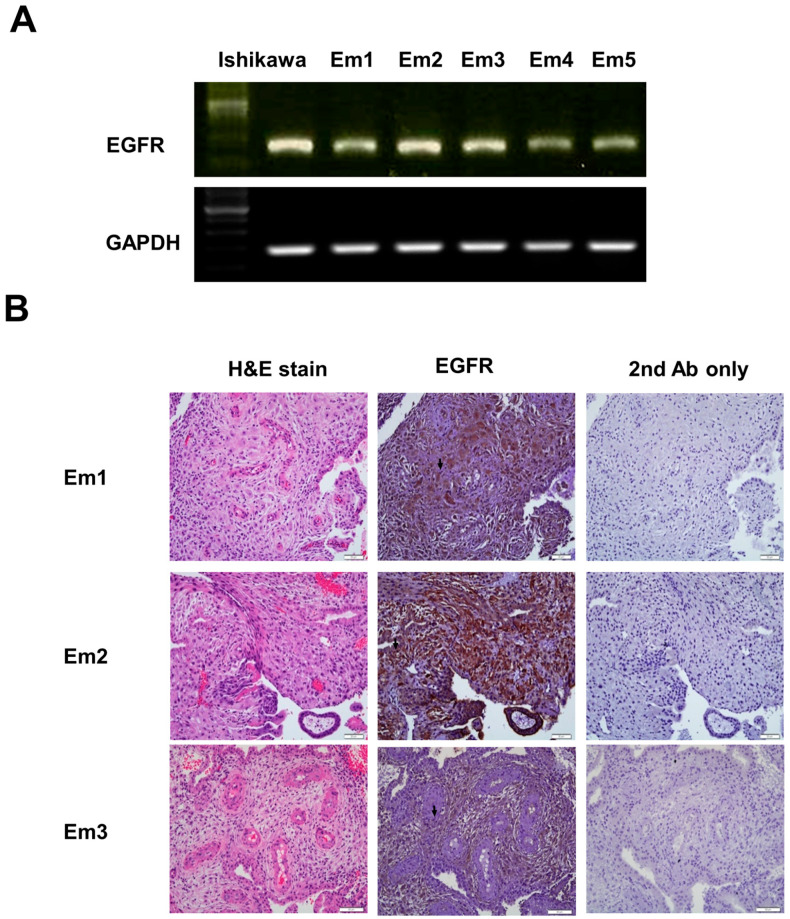
EGFR protein and mRNA expression in human decidual endometrial tissue and stromal cells. (**A**) EGFR mRNA expression in human decidual endometrial stromal cells with RT-PCR (Em 1, Em 2, Em 3, Em 4, and Em 5). Ishikawa endometrial cancer cells acted as a positive control. GAPDH was multiplied to verify the same loading. The data are representative of three individual experiments. (**B**) Immunohistochemical analysis of EGFR protein expression. Brown staining is presented in the middle of three columns describing decidual endometrium sections, including stromal cells (arrow). Sections were counterstained with hematoxylin to demonstrate the nuclei representing decidual endometrium sections in the right of three columns. Sections stained without EGFR antibody serve as a negative control in the left of three columns describing decidual endometrium sections. Micrographs were performed with a 40× objective lens. Scale bars represent 20 μm.

**Figure 4 ijms-24-15271-f004:**
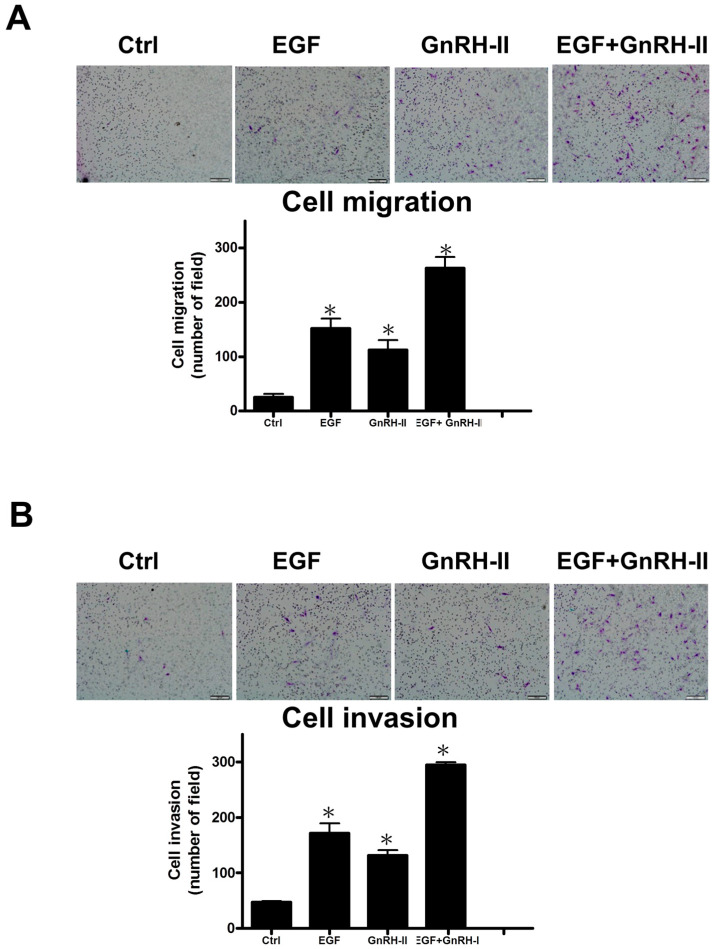
GnRH-II and EGF cotreatment to enhance the cell motility of decidual stromal cells. (**A**) Human decidual endometrial stromal cells were individually treated with 100 ng/mL of EGF, 100 nM of GnRH-II, and in combination for 24 h and then seeded by applying the Transwell migration assay. (**B**) Human decidual endometrial stromal cells were seeded onto a Matrigel-precoated filter in the Transwell chambers with 100 ng/mL of EGF, 100 nM of GnRH-II, and in combination for 48 h. We removed harvested cells in the upper side of the filter after culturing and then fixed, stained, and calculated the migrated or invaded cells. Low, representative pictures. Columns, the mean number of migrated or invaded cells of five field triplicate wells from three independent experiments; bars, SE; * *p* < 0.05, versus control. Scale bars represent 50 μm.

**Figure 5 ijms-24-15271-f005:**
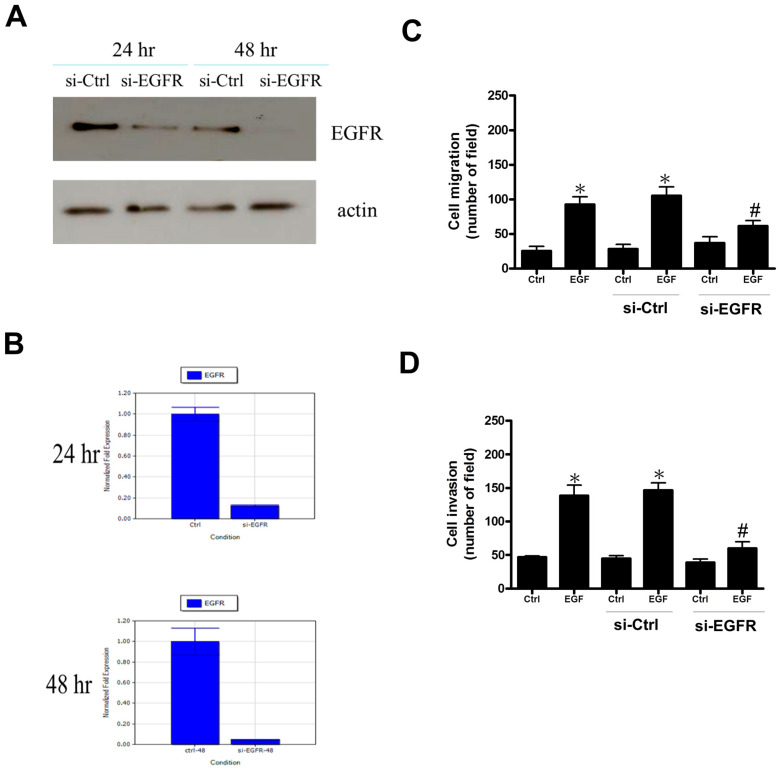
Impacts of human EGF receptor siRNA transfection on decidual stromal cells. (**A**) Immunoblot assays showed the levels of EGF receptors. The decidual stromal cells were transfected with human EGF receptor siRNA (si-EGFR) and scrambled siRNA (si-Ctrl) for 24 h and 48 h with Lipofectamine RNAiMAX. (**B**) Total RNA was isolated, and cDNA was used as normalized fold expression over the control level in RT-PCR to assess the effect of si-EGFR on EGFR mRNA expression for 24 h and 48 h. (**C**) The impacts of EGF receptor siRNA (si-EGFR) transfection on EGF-induced cell migration. Decidual stromal cells were transfected with si-EGFR and treated with EGF (100 nM) every 24 h for 72 h. We applied a migration assay to evaluate cell migration, and the results revealed that transfection with si-EGFR partially abolished EGF-induced cell migration. The data are presented as the mean ± SEM from three independent experiments (* *p* < 0.05 versus control; # *p* < 0.05 versus EGF). (**D**) The impacts of EGF receptor siRNA (si-EGFR) transfection on EGF-induced cell invasion. Decidual stromal cells were transfected with si-EGFR and treated with EGF (100 nM) every 24 h for 72 h. We performed an invasion assay to examine cell invasion, and the results showed that transfection with si-EGFR partly decreased EGF-induced cell invasion. The data are shown as the mean ± SEM from three independent experiments (* *p* < 0.05 versus control; # *p* < 0.05 versus EGF).

**Figure 6 ijms-24-15271-f006:**
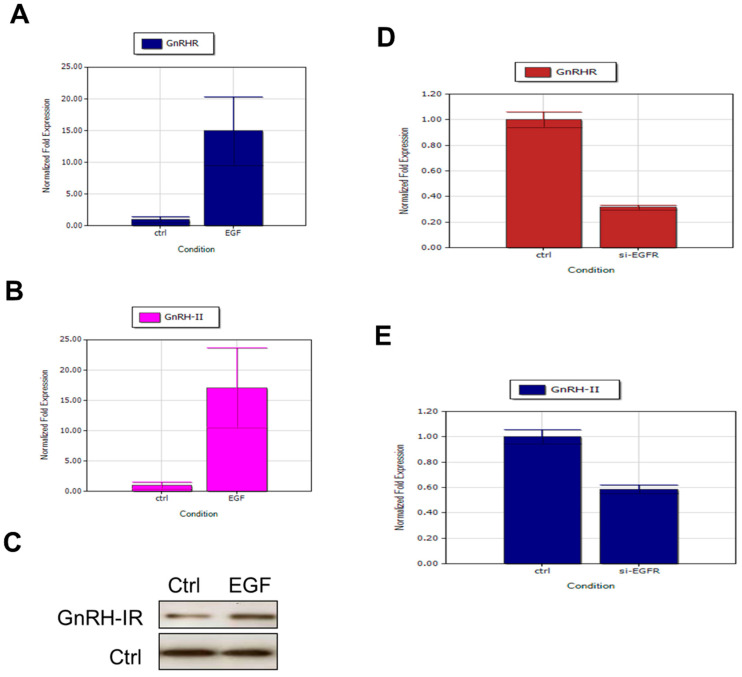
EGF increased the expression of GnRHR and GnRH-II in decidual stromal cells. Human decidual endometrial stromal cells were treated with 100 ng/mL of EGF for 24 h and then seeded using real-time PCR and immunoblot analysis. EGF-treated cells reveal increased mRNA expression of GnRH-I receptors (**A**) and GnRH-II (**B**) compared with untreated controls. (**C**) EGF-treated cells reveal increased protein expression of GnRH-I receptors compared with untreated controls. (**D**) Knockdown of endogenous EGFR with treatment with 50 nM EGFR siRNA significantly suppressed the EGF-induced mRNA expression of GnRH-I receptors and GnRH-II (**E**).

**Figure 7 ijms-24-15271-f007:**
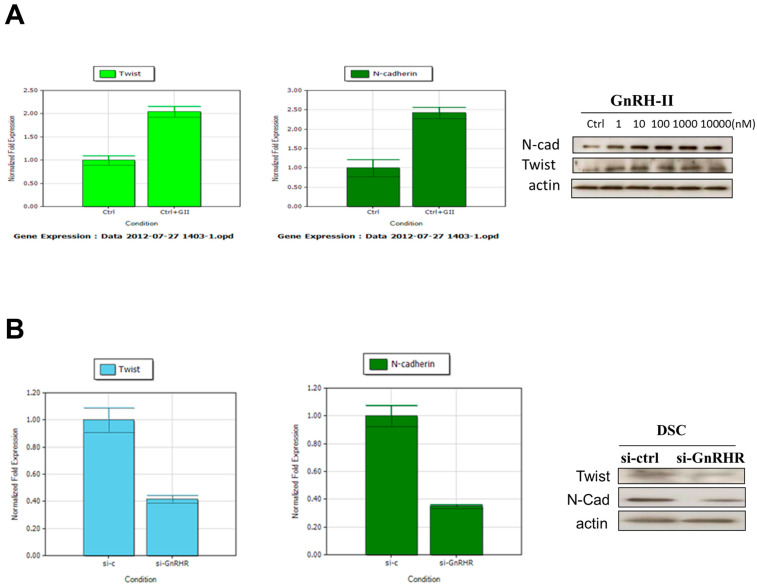
GnRH-II-induced Twist and N-cadherin mediated decidual stromal cell motility. (**A**) We applied RT-PCR and immunoblot assay to identify GnRH-II-induced Twist and N-cadherin expression in human decidual endometrial stromal cells. (**B**) We used human si-GnRH-IR or scrambled siRNA (si-Ctrl) to transfect the human decidual endometrial stromal cells with Lipofectamine RNAiMAX for one day. Knockdown of the endogenous GnRH-I receptor remarkably suppressed the mRNA and protein expression of GnRH-II-induced Twist and N-cadherin.

**Figure 8 ijms-24-15271-f008:**
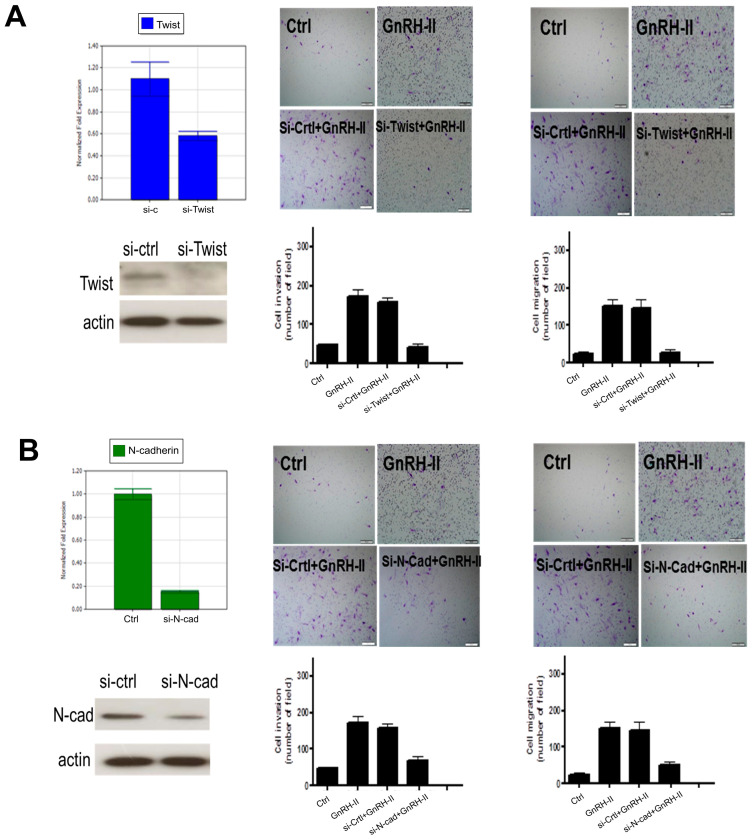
Twist and N-cadherin regulate GnRH-II-induced cell motility in human decidual endometrial stromal cells. (**A**) The impacts of si-Twist transfection on GnRH-II-stimulated cell invasion and migration. GnRH-II (1 μM) was used to treat the pre-transfected cells with si-Twist for 24 h. Invasion and migration assays were applied to evaluate cell motility. The down-regulated endogenous Twist prominently abolished GnRH-II-induced cell invasion and migration. (**B**) The impacts of si-N-cadherin transfection on GnRH-II-stimulated cell invasion and migration. GnRH-II (1 μM) was used to treat the pre-transfected cells with si-N-cadherin for 24 h. Invasion and migration assays were applied to evaluate cell motility. Knockdown of the endogenous N-cadherin significantly suppressed GnRH-II-induced cell invasion and migration. Scale bars represent 50  μm.

**Figure 9 ijms-24-15271-f009:**
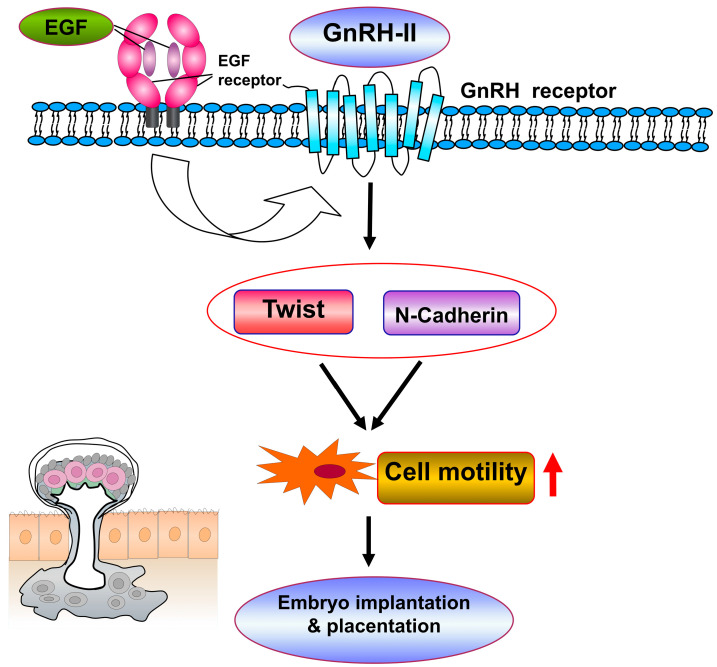
The proposed signaling pathways involved in decidual stromal cell motility are mediated by GnRH-II and EGF cotreatment. EGF acting on its receptor enhances the expression of GnRH-II and GnRH receptors. GnRH-II and EGF cotreatment regulates cell motility in human decidual stromal cells by activating Twist and N-cadherin pathways.

## Data Availability

The data that support the findings of this study are available within the article.

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
