# Peer review of "EGF-Enhanced GnRH-II Regulation in Decidual Stromal Cell Motility through Twist and N-Cadherin Signaling"

_ijms, 2023, doi:10.3390/ijms242015271_

Round 1

Reviewer 1 Report (New Reviewer)

1. What is the main question addressed by the research? The role of EGF and GnRH-II in the motility of human decidual endometrial stromal cells.

2. Do you consider the topic original or relevant in the field? Does it address a specific gap in the field? Yes, the authors showed that the co-action of EGF and GnRH-II mediated cell motility in the human decidual stromal cells.

3. What does it add to the subject area compared with other published material? -L45 the roles of EGF should be written in more detail ("role in cell biology" is too broad)

4. What specific improvements should the authors consider regarding the methodology? What further controls should be considered? -L367 what is "DSCs"? -L368 104 cells -> 10^4 cells -L419 105 cells -> 10^5 cells

5. Are the conclusions consistent with the evidence and arguments presented and do they address the main question posed? Yes, I think so.
6. Are the references appropriate? Yes.

7. Please include any additional comments on the tables and figures. -Fig.2A Is ESC "endometrial Stromal Cells"? If so, please indicate in the Figure legends. -Fig.3A the left lane seems to be marker, but it is not needed. -L139 and Fig.4 legend, about the concentration of GnRH-II, is 10 nM in L139 and 100 nM in L149&L152, which is correct? -Fig.7&8 are too small to read. Please change the graphs.

Author Response

Reviewer 2 Report (New Reviewer)

In the era of reproductive struggles, ways to improve embryo implantation are continuously scrutinized. Successful implantation requires good-quality embryos, receptive endometrium, and synchronized embryo–endometrial crosstalk. Thus far, we have the certified fact that the gonadotropin-releasing hormone pathway plays a vital role in the hypothalamus-pituitary-gonadal reproduction axis. Both GnRH and GnRH receptors are expressed not only in the hypothalamic-pituitary but also in the endometrium and embryos; their expression reaches the highest levels at the secretory-phase endometrium and at the stage of the expanded blastocyst. GnRH agonists have direct effects and functional roles in the endometrium and embryos. It enhances endometrium receptivity and embryo development. EGF was extensively studied for its role in malignancy progression. It is a bold and valuable initiative to study the combined effect of EGF and GnRH-II on the motility of decidual stromal cells and their coaction mechanisms.

I have a few minor issues to raise for the authors.

Point 1 Line 10 Define EGF

Point 2 Line 29 transforming, not adapting

Point 3 Line 66 Explain the provenance of the human decidual endometrial stromal cells as described in the Methods section.

The English language quality is good.

Author Response

This manuscript is a resubmission of an earlier submission. The following is a list of the peer review reports and author responses from that submission.

Round 1

Reviewer 1 Report

Manuscript ID: ijms-2505108

Title: EGF-enhanced GnRH-II regulation in decidual stromal cell motility through Twist and N-Cadherin signalings

Authors: Hsien-Ming Wu, Liang-Hsuan Chen , Hong-Yuan Huang , Hsin Shih Wang , Chia-Lung Tsai

In their attempt to elucidate the potential roles of the EGF/EGFR and GnRH-II in early pregnancy, the authors utilized a primary endometrial stromal culture and a variety of functional assays and expression analyses concluding that GnRH-II promoted the cell motility of human decidual endometrial stromal cells through the GnRH-I receptor and the activation of Twist and N-Cadherin signaling. Although intriguing, a major concern regarding the purity of the isolated primary endometrial stromal cell cultures would need to be resolved for this report to be scientifically valid for publication.

Major concerns:

1-    The authors need to validate the purity of the isolated primary endometrial stromal cell cultures. For instance, this could be achieved by immunocytochemical staining for vimentin, cytokeratin, muscle actin, and factor VIII. The authors need to exclude the presence of trophoblast cells in their primary endometrial stromal cell cultures to validate their data and conclusions.

2-    Many of the discussion notes were reiterated several many times in the results sections (e.g., lines 276- 288 pages 11-12). Please revise this section accordingly.

3-    The current discussion is extremely short and redundant. Please provide more insights and valid discussion notes into the potential molecular mechanisms of the EGF/EGFR/GnRH I/II and Twist/N-Cadherin in the process of regulating endometrial stromal cell motility and embryo implantation.

4-    Results section: Figure 6 D and E: please provide gel images documenting the differences in cDNA copies of the GnRHR and GnRH II with and without the addition of the si-EGFR accordingly.

5-    Results section: Figure 7 needs to be broken up into several figures- Currently Figure 7 is too crowded with data from expression analyses and migration and invasions assays. Please revise accordingly.

6-    Figure 8 is a conclusion rather than a result. Please consider removing current Figure 8 from the results section.

Minor concerns:

1-    Please provide appropriate citations to support your arguments in lines 243-244, 264, and 271-272 on page 11.  

2-    Please replace the word “generates” in line 253 p11 with the word “modulates”.

3-    Please add the word “cells” at the end of the sentence on line 257.

4-    The sentence on lines 258/259 p11 does not make sense. Please consider re-phrasing.

Multiple grammar errors were found throughout the text of the manuscript. Please check and re-check for the English grammar of your manuscript. 

Author Response

Responses to the comments:

Reviewer 1

  1. The authors need to validate the purity of the isolated primary endometrial stromal cell cultures. For instance, this could be achieved by immunocytochemical staining for vimentin, cytokeratin, muscle actin, and factor VIII. The authors need to exclude the presence of trophoblast cells in their primary endometrial stromal cell cultures to validate their data and conclusions.

    Ans : We revised the description as suggestion in the cell culture section of Materials and Methods. We cited the previous reports and performed the modified protocol to collect and purify human decidual stromal cells from decidual tissue through enzymatic digestion and mechanical dissociation.

  1. Many of the discussion notes were reiterated several many times in the results sections (e.g., lines 276- 288 pages 11-12). Please revise this section accordingly.

Ans : We revised the description as suggestion.

  1. The current discussion is extremely short and redundant. Please provide more insights and valid discussion notes into the potential molecular mechanisms of the EGF/EGFR/GnRH I/II and Twist/N-Cadherin in the process of regulating endometrial stromal cell motility and embryo implantation.

Ans : We revised the description as suggestion.

  1. Results section: Figure 6 D and E: please provide gel images documenting the differences in cDNA copies of the GnRHR and GnRH II with and without the addition of the si-EGFR accordingly.

Ans : In Figure 6 D and E, Real-time PCR was performed by using Complementary DNA (cDNA) synthesized from RNA samples. The figure demonstrated the normalized fold expression of the EGF-induced GnRH-I receptors and GnRH-II.

  1. Results section: Figure 7 needs to be broken up into several figures- Currently Figure 7 is too crowded with data from expression analyses and migration and invasions assays. Please revise accordingly.

Ans : We revised the Figures and description as suggestion.

  1. Figure 8 is a conclusion rather than a result. Please consider removing current Figure 8 from the results section.

Ans : We revised the description as suggestion.

  1. Please provide appropriate citations to support your arguments in lines 243-244, 264, and 271-272 on page 11.

Ans : We revised the description as suggestion.

  1. Please replace the word “generates” in line 253 p11 with the word “modulates”.

Ans : We revised the description as suggestion.

  1. Please add the word “cells” at the end of the sentence on line 257.

Ans : We revised the description as suggestion.

  1. The sentence on lines 258/259 p11 does not make sense. Please consider re-phrasing.

Ans : We revised the description as suggestion.

  1. Multiple grammar errors were found throughout the text of the manuscript. Please check and re-check for the English grammar of your manuscript.

Ans : We revised the description as suggestion.

Reviewer 2 Report

In the present study, the authors of the manuscript entitled: "EGF-enhanced GnRH-II regulation in decidual stromal cell motility through Twist and N-Cadherin signalings" demonstrated in detail and at a high methodological level the involvement of EGF and GnRH-II through acting on EGFR and GnRH-I receptors, respectively, in the migration and invasion of decidual endometrial stromal cells, which play an important role in early pregnancy. In addition, the authors demonstrated that EGF-treated cells increased mRNA and protein expression of GnRH-I receptor compared to untreated controls. In turn, the GnRH-II-induced expression of Twist and N-Cadherin in human decidual endometrial stromal cells was mediated by GnRH-I receptor. The transactivation of epidermal growth factor receptor (EGFR), a subfamily of RTKs, by GPCRs has been already demonstrated [Wang Z. Transactivation of Epidermal Growth Factor Receptor by G Protein-Coupled Receptors: Recent Progress, Challenges and Future Research. Int J Mol Sci. 2016 Jan 12;17(1):95. doi: 10.3390/ijms17010095. PMID: 26771606; PMCID: PMC4730337]. In the present study, the authors demonstrated the reverse effect of epidermal growth factor receptor (EGFR) on GPCR (GnRH-I receptor). Despite the high quality of the data obtained, there are several significant shortcomings of this article.

Major revisions:

1)     In the section “Introduction”, it is necessary to detail the molecular mechanisms of regulation of embryo implantation: what is known and what are the gaps in knowledge. Therefore, it is not clear from the Introduction why it was necessary to analyze the co-action of EGF and GnRH-II on the stromal cells of the decidua. Why were Twist and N-cadherin chosen to assess the effect of EGF and GnRH-II on migration and invasion of stromal cells?

2)     Section 2.2. The authors of the article wrote: “To identify then GnRH-IR expression, we extracted the primary culture of human decidual endometrial stromal cells and investigate the GnRH-IR expression by immunoblot analysis and immunohistochemical studies.” But the results of immunoblot and immunohistochemistry were not presented. Only references to previous studies are provided.

3)     Most of the references to literary sources are 20-30 years old. It is necessary to analyze the participation of EGF and GnRH-II in embryo implantation in terms of the data obtained to date.

4)     The discussion partially repeats the introduction without any new semantic load.

Minor revisions:

1)      The word "role" is missing from the sentence:Additionally, epidermal growth factor (EGF), an autocrine growth factor, plays a key «role» in cell biology and 41 is a critical therapeutic target in reproductive cells [9,10]

2)     Figure 1A: Misspelled 1µM

3)     Figure 3B: The authors wrote: “Brown staining was presented in the middle of three columns describing decidual endometrium sections, including stromal cells (arrow)”, but there is no arrow in the image.

4)     Figure 5A: Misspelled word “si-Ctrl”.

Moderate editing of English language required

Author Response

Reviewer 2

  1. In the section “Introduction”, it is necessary to detail the molecular mechanisms of regulation of embryo implantation: what is known and what are the gaps in knowledge. Therefore, it is not clear from the Introduction why it was necessary to analyze the co-action of EGF and GnRH-II on the stromal cells of the decidua. Why were Twist and N-cadherin chosen to assess the effect of EGF and GnRH-II on migration and invasion of stromal cells?

Ans : We revised the description as suggestion.

  1. Section 2.The authors of the article wrote: “To identify then GnRH-IR expression, we extracted the primary culture of human decidual endometrial stromal cells and investigate the GnRH-IR expression by immunoblot analysis and immunohistochemical studies.” But the results of immunoblot and immunohistochemistry were not presented. Only references to previous studies are provided.

Ans : We revised the description as suggestion.

  1. Most of the references to literary sources are 20-30 years old. It is necessary to analyze the participation of EGF and GnRH-II in embryo implantation in terms of the data obtained to date.

Ans : We revised the description as suggestion.

  1. The discussion partially repeats the introduction without any new semantic load.

Ans : We revised the description as suggestion.

  1. The word "role" is missing from the sentence:Additionally, epidermal growth factor (EGF), an autocrine growth factor, plays a key «role» in cell biology and 41 is a critical therapeutic target in reproductive cells [9,10]

Ans : We revised the description as suggestion.

  1. Figure 1A: Misspelled 1µM

Ans : We revised the description as suggestion.

  1. Figure 3B: The authors wrote: “Brown staining was presented in the middle of three columns describing decidual endometrium sections, including stromal cells (arrow)”, but there is no arrow in the image.

Ans : We revised the description as suggestion.

  1. Figure 5A: Misspelled word “si-Ctrl”.

Ans : We revised the description as suggestion.

Round 2

Reviewer 1 Report

Major concerns:

1- The authors presented outdated data in Figure 7A- these data show a time-stamp of 2012-07-27! The authors must justify their use of such outdated data to support their current research, otherwise, this is an ethically unacceptable academic and research practice. 

2- The authors failed to adequately respond to my previous concerns over the purity of their primary cell culture! It is never enough, nor it is scientifically adequate to merely cite previous work in lieu of clearly demonstrating cellular identifiers of their primary endometrial stromal cell cultures to support the authors' conclusions.  

Multiple English grammar errors were detected in the revised version of this manuscript. Many sentences are grammatically wrong and need to be re-phrased/re-written adequately (e.g., "Furthermore, the impact of EGFstimulated GnRH-II expression develops a particular autocrine/paracrine regulation associated with gynecological cell motility"))!  

Reviewer 2 Report

The authors of the manuscript made all the recommended corrections. It is proposed to accept a new version of the manuscript for publication.